# Implementing Precision Medicine in Human Frailty through Epigenetic Biomarkers

**DOI:** 10.3390/ijerph18041883

**Published:** 2021-02-15

**Authors:** José Luis García-Giménez, Salvador Mena-Molla, Francisco José Tarazona-Santabalbina, Jose Viña, Mari Carmen Gomez-Cabrera, Federico V. Pallardó

**Affiliations:** 1U733, Centre for Biomedical Network Research on Rare Diseases (CIBERER-ISCIII), 28029 Madrid, Spain; j.luis.garcia@uv.es (J.L.G.-G.); Federico.V.Pallardo@uv.es (F.V.P.); 2Mixed Unit for Rare Diseases INCLIVA-CIPF, INCLIVA Health Research Institute, 46010 Valencia, Spain; 3Department of Physiology, Faculty of Medicine, University of Valencia, 46003 Valencia, Spain; salva.mena@gmail.com; 4EpiDisease S.L., Parc Cientific de la Universitat de València, 46980 Paterna, Spain; 5Servicio de Geriatría, Hospital Universitario de la Ribera, CIBERFES, Alzira, 46010 Valencia, Spain; fjtarazonas@gmail.com; 6Freshage Research Group, Department of Physiology, Faculty of Medicine, Institute of Health Research-INCLIVA, University of Valencia and CIBERFES, 46010 Valencia, Spain; jose.vina@uv.es

**Keywords:** geriatric syndromes, healthy aging, exercise, histones, DNA methylation, non-coding RNA

## Abstract

The main epigenetic features in aging are: reduced bulk levels of core histones, altered pattern of histone post-translational modifications, changes in the pattern of DNA methylation, replacement of canonical histones with histone variants, and altered expression of non-coding RNA. The identification of epigenetic mechanisms may contribute to the early detection of age-associated subclinical changes or deficits at the molecular and/or cellular level, to predict the development of frailty, or even more interestingly, to improve health trajectories in older adults. Frailty reflects a state of increased vulnerability to stressors as a result of decreased physiologic reserves, and even dysregulation of multiple physiologic systems leading to adverse health outcomes for individuals of the same chronological age. A key approach to overcome the challenges of frailty is the development of biomarkers to improve early diagnostic accuracy and to predict trajectories in older individuals. The identification of epigenetic biomarkers of frailty could provide important support for the clinical diagnosis of frailty, or more specifically, to the evaluation of its associated risks. Interventional studies aimed at delaying the onset of frailty and the functional alterations associated with it, would also undoubtedly benefit from the identification of frailty biomarkers. Specific to the article yet reasonably common within the subject discipline.

## 1. Introduction

The concept of frailty has been evolving for more than 20 years. Since the publication of a validated phenotype of frailty as a medical syndrome in 2001 by Fried and colleagues [1], this geriatric condition has received growing interest due to its association with longevity and aging-related phenotypes (Figure 1). At the moment there is no consensus on the definition of frailty, but it is accepted that frailty reflects a state of increased vulnerability to stressors as a result of decreased physiologic reserves, and even dysregulation of multiple physiologic systems leading to adverse health outcomes for individuals of the same chronological age [2]. From a gerontological point of view, frailty is a stochastic, deleterious and dynamic process of deficit accumulation. Cellular deficits include: senescence and stem cell exhaustion, loss of proteostasis, decline in metabolism, inflammation, DNA damage and deficit in DNA repair, hormone dysregulation, and epigenetic alterations [3]. The accumulation of these deficits varies across life stages and some individuals are more predisposed to them [4]. Since cells are the primary sites of deficit accumulation, cellular frailty may be the major driver of the systemic physiological decline of tissues and organs [5], which may lead to late-onset multimorbidity.

Frailty is the main determinant of longevity and quality of life in the elderly population and it has become a public health concern [6]. Among the causes of frailty, one of most important contributors is the adoption of unhealthy lifestyles (i.e., physical inactivity and/or sedentarism, malnutrition, smoking and alcohol intake) [7]. Frailty is a dynamic process, characterized by frequent transitions between states of frailty and towards disability and dependency. Disability is characterized by a functional limitation that increases the demand for long-term care services for the elderly, which infers high social-health and personal costs [8]. Frailty increases dramatically with age, with a prevalence of 5.2% in men and 9.6% in women over the age of 65 years [9]. These figures grow to 40% in adults aged 80 years and older. Frailty increases the risk of falls, delirium, disability and other geriatric syndromes [10,11]. It also increases vulnerability to age-related disorders, such as myocardial infarction, diabetes and hypertension in those who suffer from it [12].

In the past two decades, a large proportion of the global burden of disease has changed due to high rates of disability resulting from morbidity of non-communicable diseases. In fact, it is considered that disability increases health costs more than disease by itself [2]. In this scenario, we have to define phenotypic aging, which includes the changes in body composition and structures (i.e., loss of skeletal muscle mass), energetics, homeostatic mechanisms, and neuronal function that occur while we age and that may contribute to functional aging. Functional aging is referred to as the age-associated decline in physical, emotional, cognitive, and social functions leading to a decrease in the performance of basic activities of daily living and contributing to the loss of independence [13].

Recent research is helping to shed light on mechanisms underlying frailty, how frailty can influence disease onset and progression, and how new interventions can attenuate frailty to improve health status. Moreover, frailty helps to explain heterogeneity in aged people and it provides the basis to understand the differences between biological and chronological age.

The main physiological systems dysregulated in the frail patient include the endocrine [14], musculoskeletal [15], respiratory [16], renal, cardiovascular [17], immune [18], hematopoietic system [19], and also the nervous system [20]. Moreover, mounting evidence indicates that frailty may increase the risk of mild cognitive impairment and contribute to dementia [21]. This probably occurs because frailty and cognitive disorders may share common biological pathways. Thus, frail older adults may be at higher risk of incident cognitive disorders than robust ones.

Several predictive models related to frailty are available in the literature. Most of these predictive models are focused on defining and validating a frailty index, which is used as a predictor of disability, hospitalization, and mortality. There are also a smaller number of studies in which these indexes are used to predict the ability to perform activities of daily living (ADL), instrumental activities of daily living (IADL), or the risk of falls [22]. In any case, each of these indexes can improve the clinical management of frailty.

A key approach to overcoming the challenge of frailty is to implement precision medicine by using biological biomarkers to improve diagnostic accuracy and to optimize its management. Finding biomarkers would allow gerontologists to predict the functional trajectories of older adults at preclinical stages. This could help to develop early interventions aimed at preventing frailty and its natural progression to disability. Precision medicine was defined by the National Research Council’s Toward Precision Medicine in 2008 as: “The tailoring of medical treatment to the individual characteristics of each patient … to classify individuals into subpopulations that differ in their susceptibility to a particular disease or their response to a specific treatment. Preventive and therapeutic interventions can then be concentrated on those patients who will benefit, sparing expense and side effects for those who will not”.

Today we know, thanks to the “Human Genome Project” and the “Human Epigenome and the Human Epigenome Roadmap Projects”, that genetics is not the only contributor to disease. In most complex diseases and human conditions genetics cannot, by itself, explain the deficits or molecular alterations related to the onset of the disease, its progression or even the response to a specific treatment. In fact, most human diseases are complex multifactorial pathologies, caused by genetic background and epigenetic inputs, which can modulate transcriptional programs and lead to adverse clinical outcomes.

Epigenetics is defined as the discipline that studies the regulation of gene expression by mechanisms not related to changes in the DNA sequence. These regulatory processes can be heritable and set the features of specific cell lineages and subpopulations. Many of these epigenetic mechanisms suffer the influence of environmental factors and are part of the adaptive homeostatic mechanisms of all organisms. Thus, the epigenetic regulatory mechanisms are continuously being implemented for surviving. Organisms with powerful adaptive mechanisms have developed an extremely complex epigenetic machinery and if the organism has a long-life span, like in humans, the epigenetic regulatory actions are expected to play a very important role.

Epigenetics is a rising discipline in biomedicine, which aims to improve predictive and precision medicine by discovering new mechanisms underlying diseases and providing new biomarkers in order to identify molecular targets that are modulable, for instance, by using epigenetic drugs [23,24,25]. Most human biological processes have complex multifactorial modulators that include polymorphisms and copy number variation in human genes, besides epigenetic mechanisms, that contribute to the modulation of gene expression [26].

According to Pal and Tyler [27] the main epigenetic features in aging are: (i) reduced bulk levels of core histones, (ii) altered pattern of histone post-translational modifications, (iii) changes in the pattern of DNA methylation, (iv) replacement of canonical histones with histone variants, and (v) altered expression of non-coding RNA. The consequence of these changes in the epigenetic regulation alters the local accessibility of the transcriptional and DNA repairing machinery to the genetic material, thus, producing among others, aberrant gene expression, reactivation of transposable elements and genomic instability [27]. Epigenetics is able to explain, in most cases, the importance of life style factors and the influence of the environment on the aging process.

We have defined an epigenetic biomarker as “any epigenetic mark or altered epigenetic mechanism which generally serves to evaluate health or disease status and is particularly stable and reproducible during sample processing and analysis” [25]. There is a current need to better understand the etiology of frailty to develop effective interventions for its prevention, amelioration, or even its reversion. The identification of potential sensitive and specific biomarkers may provide insights into the molecular, metabolic, cellular, and physiological alterations that lead human beings to frailty [28].

The present review aims to provide answers to some of the uncertainties regarding the role of epigenetic mechanisms in frailty development.

## 2. Materials and Methods

The present review was carried out by conducting an electronic search in OVID (Medline and Embase), combining the following MeSH keywords: “epigenomics” or “biomarkers”, combined with “frailty”. The search was limited to publications in the last ten years, in English, Spanish, and French. A total of 158 articles were obtained of which 87 were finally selected. The MeSH construction (“Epigenomics” [Mesh] OR “Epigenetic Repression” [Mesh] OR “Epigenesis, Genetic” [Mesh] OR “Biomarkers” [Mesh]) AND “Frailty” [Mesh] were used. Some additional instructions were added for certain specific objectives where necessary. In 14 cases, supplementary information was obtained in the form of references of the selected articles.

The articles were selected by the investigators based on the following inclusion criteria: randomized clinical trials, cohort studies, case-control studies, observational studies, and before and after analyses; population: older adults; outcome geriatric syndromes and frailty. The exclusion criteria were: letters to the editor, case reports, manuscripts with no available abstract or those with only the abstract published. All the published studies were re-evaluated by the authors of the review, and final inclusion was restricted to those of sufficient quality to afford information pertinent to the objectives of this review.

## 3. Results

Frailty has become a public health concern. A key approach to overcoming the challenge of frailty is to implement precision medicine by using biological biomarkers to improve diagnostic accuracy and to optimize its management. Finding epigenetic biomarkers would provide an excellent tool to predict the evolution of frailty to disability and dependency in older individuals as well as contribute to designing personalized pharmacological and non-pharmacological interventions.

### 3.1. Epigenetic Mechanisms Related to Aging and Frailty

Importantly, changes across cell generations accumulate genetic and epigenetic alterations [29]. Among the different generalized changes during aging in mammals, global DNA hypomethylation mainly in DNA repeats, as well as, local hypermethylation at specific gene promoters have been found [30]. These can result in gene expression silencing in DNA repair and anti-inflammatory genes [31]. It has also been suggested that other genes participating in the maintenance of muscle and nervous systems processes, chromatin remodeling, and transcription control may also be affected [32]. It has been proposed that these global changes in DNA methylation are associated with the incomplete restoration of epigenetic patterns after DNA replication or DNA repair [33], which contribute to the accumulation of epimutations in the bulk DNA. These changes in the methylation pattern together with other epigenetic related factors associated with aging were coined the “epigenetic drift [34]” by Veitia and coworkers since it was considered as a general and progressive epigenetic process directly related to senescence, which is especially important in tissue-specific stem cells. Among these adult stem cells, muscle satellite cells play a very important role in sarcopenia, which is one of the key features in frailty. The important changes that take place in satellite cells upon aging have been recently demonstrated in mouse muscle stem cells [35], underscoring the importance of skeletal muscle epigenetic regulation in the development of the frail phenotype.

#### 3.1.1. DNA Methylation

Since its discovery, DNA methylation (DNAm) has been the best studied epigenetic modification. It affects the 5′ carbon position of cytosine, mostly in the context of CpG dinucleotide. There is a significant amount of work describing the analysis of DNA methylation and its role in human diseases. From a clinical point of view, DNA methylation harbors great potential value as a diagnostic and prognostic biomarker. The possibility of screening blood-circulating DNA or DNA extracted from leukocytes for alterations in DNA methylation also increases its value as an epigenetic biomarker.

It has been demonstrated that DNA methylation displays a strong correlation with age and age-related processes. In a pioneering work published in PNAS in 2005, global and locus-specific differences in DNA methylation in identical twins of various ages was found. Moreover, the authors demonstrated a correlation between DNA methylation and quality of life during aging and also that DNA methylation can be influenced by different environmental and lifestyle factors [33]. Seven years later, using DNA from peripheral blood cells, an association between global DNA methylation levels and age-related functional decline was reported [36]. The authors found lower global methylation levels in frail subjects than in those who had a better functional status such as pre-frail and non-frail [37]. This finding was accompanied by a longitudinal study to explore whether the methylation levels were sensitive to changes in the frailty status over time. It was found that the decrease in global DNA methylation was associated with functional decline and not to chronological age, at least after 65 years of age [36]. This led to the suggestion that biological age and aging acceleration can be predicted based on methylation patterns at specific CpG sites [38].

Epigenetic clocks, which are based on the use of specific DNA methylation patterns, can also be correlated to individuals’ chronological ages to assess inter-individual and/or inter-tissue variability in the aging rate [38]. For a number of epigenetic clocks, the divergence between epigenetic age and chronological age reflects biological age acceleration. In this regard, epigenetic age acceleration has been associated with the risk of heart disease [39], breast cancer [40,41], lung cancer [42], neurodegenerative diseases such as Alzheimer’s disease [43], and it has also been associated with differential susceptibility to death [44,45,46,47].

There are up to eleven epigenetic clocks [48], which incorporate a subset of CpGs (between 1 and 513) that are differentially weighted to estimate epigenetic age and predict age-related outcomes.

The most validated epigenetic clock was proposed by Horvath (the Horvath’s clock) [40]. It uses the exact 353 CpG loci predictor to analyze DNAm age acceleration. Other epigenetic clocks are also gaining relevance, such as the Hannum’s clock [49] or the clock proposed by Zhang and co-workers that was developed in 2017 as a predictor for all-cause mortality [47]. More recently, an epigenetic clock for lifespan and health span has been developed to differentiate same-aged individuals based on morbidity and mortality risk [45].

Over the years, additional approaches analyzing a lower number of CpG sites for an estimation of the epigenetic age have also been proposed [50,51]. Two clocks have been developed by Wolfgang Wagner’s group: the 3 CpG model [50] and the 99 CpG model [52]. Weidner and co-workers selected 3 CpGs from DNAm in blood from array data of the 27K Illumina array, which were later validated by Lin and Wagner using the Infinium 450K Beadchip array (Illumina) [52]. This resulted in an age predictor with an average accuracy of 5.4 years. Vidal-Bralo et al. [51] developed an epigenetic clock analyzing the DNAm in 390 healthy subjects. This clock consists of an age predictor based on 8 CpG sites, out of a preselected list of the most informative CpGs (with an age correlation over 0.85). More recently, Liu and co-workers have also proposed a “Meta-Clock”, consisting of a new approach by deconstructing the clocks into submodules and recombining them into a more robust epigenetic aging measurement [48]. This clock has demonstrated improved prediction for mortality and more robust aging associations with DNAm [48].

The influence of genetic and environmental factors as well as technical differences in the selected DNAm analysis methods have been highlighted as the main drawbacks of the age-related CpGs analysis [53].

Several studies have shown that epigenetic age acceleration is associated with frailty in older individuals [54,55]. Two laboratories have studied DNA methylation changes in frailty using the deficit accumulation method [56] and the frailty phenotype [55].

#### 3.1.2. Histone Post-Translational Modifications and Histone Variants

Chromatin is organized by repeating arrays of nucleosomes, which consist of 145 bp of DNA wrapped around a histone octamer. Each histone octamer consists of two copies of each histone, H4, H3, H2A, and H2B. Histones can be chemically modified, producing histone post-translational modifications (PTMs) (i.e., acetylation, methylation, phosphorylation, butyrylation, hydroxybutyrylation, crotonylation, citrullination, formylation, glycosylation, O-GlcNAcylation, carbonylation, parsylation, and glutathionylation) [57]. These changes can modify the nucleosome structure and accessibility to different regions of the genome. Proper coordination between components of the epigenetic machineries is responsible for the introduction or removal of PTMs, which are essential for the correct control of epigenome function. Otherwise, mutations in epigenetic enzymes (i.e., writers and erasers) and/or in epigenetic readers, which alter the “histone code”, can be translated into diseases.

Histone post-translational modifications are more difficult to map completely over the whole epigenome than DNA methylation, which makes them more difficult to analyze using high throughput technologies. However, several efforts have been made in order to explore histone metabolism, histone replacement, and histone PTMs introduction and removal. In this regard, Benayoun and co-workers have described reduced bulk levels of core histones, altered histone posttranslational modifications and cellular misregulation in the replacement of histone variants in aging [58].

In different histone post-translational modifications, phosphorylation of Ser139 in the histone variant H2AX, also known as γH2AX, is a key event that marks double strand breaks in the DNA [59], and therefore, DNA instability and DNA repair deficits. Importantly, elevated levels and persistence in time of γH2AX is considered an unrepairable double strand break [60], which may result in tumorigenesis, mitotic arrest or cell death.

#### 3.1.3. Non-Coding RNA

Epigenetic regulation is also mediated by non-coding RNAs (ncRNAs), which are molecules that, despite being non-protein-coding RNAs, have important regulatory functions in gene expression. Generally, ncRNAs regulate gene expression at the transcriptional and post-transcriptional level. These ncRNAs include microRNAs (miRNAs) and long non-coding RNAs (lncRNAs). miRNAs are a large family of short RNA molecules with an average size of 17–25 nucleotides [61]. miRNAs can down-regulate gene expression by inhibiting mRNA translation when binding to its 3′-UTR or by degrading mRNA molecules, as well as increase mRNA translation of some targets [62] when miRNAs bind to 5′-UTR regions of the target mRNA. lncRNA transcripts are longer than 200 nucleotides and poorly conserved. lncRNAs are transcribed from all over the genome including intergenic regions, domains overlapping one or more exons of another transcript on the same strand (sense) or on the opposite strand (antisense), or intronic regions of protein-coding genes [63]. Many lncRNAs act by forming complexes with chromatin-modifying proteins and recruiting them to specific sites in the genome, thereby modifying chromatin states and influencing gene expression [64].

MiRNAs have gained relevance in recent years in research on aging due to their role in the control of several biological processes [65]. For instance, it has been suggested that miRNAs can control muscle metabolism and muscle wasting [66]. This opens up new avenues at different levels, such as the identification of the molecular targets to test miRNA-based interventions as a therapeutic strategy against sarcopenia [66].

Smith-Vikos and co-workers have proposed the analysis of circulating miRNAs in serum and plasma to screen for biomarkers of healthy aging and longevity [67]. Exosomes, small cell-derived vesicles found within extracellular fluids and originated from cellular multivesicular bodies fused with the plasma membrane, have also been identified as valuable biomarkers for a number of disease conditions. Exosomes can contain microRNA [68]. Inflamma-miRs, mitomiRs and myomiRs have been closely related with frailty [69] as we describe in the following section.

### 3.2. Epigenetic Biomarkers Associated with Aging and Frailty

A key approach to overcome the challenges of frailty is the development of biomarkers to improve early diagnostic accuracy and to predict clinical trajectories in older individuals. Furthermore, frailty biomarkers should allow us to stratify patients and distinguish those who will benefit from a specific therapeutic approach compared to those who will not benefit from a specific intervention, thus making frailty identification one of the pillars in decision-making in regard to older patients.

Epigenetic biomarkers should have the ability to detect early subclinical changes or deficits at the molecular and/or cellular level [70]. So, they can help to identify key dysregulated transcriptional programs, which may help to identify molecular targets for pharmacological and non-pharmacological interventions to prevent or delay the development of frailty, and also its consequences. Clinical validation of a candidate biomarker (Table 1) will provide support for the clinical diagnosis and monitoring of frailty or any of its associated risks.

It has been demonstrated that a biological epigenetic clock is better associated with frailty index [71] than the telomere length [54]. The DNA methylation patterns play a relevant role in explaining inter-individual differences in biological aging and frailty [36,72]. These studies suggest that an “accelerated” biological aging determined by epigenetic changes may be closely correlated to clinically relevant features related to frailty phenotypes [73].

#### 3.2.1. DNA Methylation as a Frailty Biomarker 

The first study performed to assess the potential relationship between epigenetic age acceleration and frailty-related characteristics was published in 2015. The authors performed a DNA methylation analysis in the Lothian Birth Cohort 1936 (LBC1936) and found significant correlation coefficients ranging from −0.05 to −0.07 between age acceleration based on the DNAm analysis and cognitive function, grip strength, or lung function [46].

Afterwards, Breitling and co-workers, studied the epigenetic clock developed by Horvath [40] in a cross-sectional observational study with 1820 subjects from two large subsets of community-dwelling older adults in the German ESTHER cohort study. These authors found that epigenetic age acceleration was correlated with clinically relevant aging-related phenotypes. More specifically, their results suggested one added functional deficit per 12 years of methylation age acceleration [54]. Importantly, in this study, the authors also found that telomere length measured in leukocytes, a well-studied parameter related with aging, was not associated with frailty index [54].

More recent studies have also explored the association between telomere length and DNAm with frailty. In the Berlin Aging Study II (BASE-II), a marginal association of age acceleration was found between leukocyte telomere length (rLTL) and DNAm [74]. However, in a recent work by Demuth et al., it was shown that neither DNAm age acceleration [51] nor rLTL were significantly associated with the Fried’s Frailty Score or the functional assessments in the Berlin Aging Study II [75]. Interestingly, only one of the analyzed assessments, the clock drawing test, was significantly associated with DNAm age acceleration in older men, with an average of 1.9 years higher DNAm age acceleration [75]. This result seems important, because the clock drawing test is used to predict dementia and detect early-stage Alzheimer’s disease [76]. The results reported by Demuth and co-workers seem contradictory to those described by Breitling’s research group [54]. However, this might be the result of the different approaches used to measure frailty index and also the different methodologies used to measure DNAm age acceleration, the 353 CpG loci measurement according to the Hovarth’s clock [40], and the seven-CpGs age acceleration method proposed by Vidal-Bralo and collaborators [51].

Further studies exploring the association of the “Meta-Clock” with frailty are needed to demonstrate its utility in assessing the relationship between epigenetic age acceleration and frailty-related phenotypes [48].

#### 3.2.2. Histone PTMs and Histone Variants as Frailty Biomarkers

The implications of increased and decreased levels of histone acetylation in enhancing and constraining cognitive functions, particularly learning and memory have been demonstrated [77]. Accordingly, several histone deacetylase inhibitors (HDACis) have proven successful in rescuing cognitive deficits in animal models of neurodegeneration and cognitive decline such as Alzheimer’s disease [77]. They might constitute a new strategy for pharmacological interventions against cognitive impairments by improving learning capacity, activating learning genes and mediating “cognitive epigenetic priming” [78]. “Cognitive epigenetic priming” is a theory that aims to explain the potential of histone acetylation to promote memory by facilitating the expression of neuroplasticity-related genes [78]. This is of special relevance, since cognitive impairment seems to be closely related with frailty, as we described above. Histone acetylation is closely related with other features associated with frailty such as sarcopenia. Global histone H3 methylation and acetylation decreases in muscle tissue with age, which may be linked to the well-known age-related type IIb fiber atrophy in skeletal muscle [79]. Walsh and collaborators demonstrated that the use of butyrate, an HDACi, increases the histone acetylation levels in skeletal muscle and prevents age-associated hindlimb muscle loss in female C57Bl/6 mice [80].

Measurement of the H2A histone family member X (H2AX) phosphorylation, at the amino acid Ser139, can provide information regarding frailty severity [81]. This was demonstrated in leukocytes and monocytes isolated from individuals classified as non-frail, pre-frail or frail depending on the Fried’s frailty score [81]. The authors found that the percentage of γH2AX in the cells and the number of positive frailty criteria were significantly correlated (r = 0.201, *p* < 0.01). In addition, they performed a multivariate statistical analysis, adjusting by gender, age, and tobacco consumption (and alternatively adjusting by BMI), and confirmed previous results from univariate analyses on the influence of frailty. Therefore, frailty severity was accompanied by a progressive decrease in DNA repair capacity in lymphocytes (*p* < 0.05). Interestingly, the authors also independently studied the association of γH2AX with each one of the five Fried’s frailty criteria: (i) unintentional weight loss; (ii) muscular weakness (grip strength); (iii) self-reported exhaustion; (iv) slow walking; and (v) low physical activity level [1] and found significantly higher γH2AX values in individuals positive for the low physical activity (*p* < 0.001), slow waking (*p* < 0.01) and low grip strength (*p* < 0.01) criteria. The main conclusion from this study is that the levels of γH2AX increase progressively according to frailty severity and that the use of the γH2AX levels, besides the micronucleus frequency in lymphocytes, could be a useful parameter to identify pre-frail and frail individuals [81]

#### 3.2.3. Non-Coding RNAs as Frailty Biomarkers

Ipson and co-workers compared the differential expression of exosome-derived miRNAs from young adults by using small RNA sequencing (smallRNA-seq) in robust and frail individuals and identified eight enriched miRNAs associated with frailty: miR-10a-3p, miR-92a-3p, miR-185–3p, miR-194–5p, miR-326, miR-532–5p, miR-576–5p, and miR-760 [82].

Rusanova and collaborators [83] explored the expression of several miRNAs by RT-qPCR and found that robust subjects had higher expression of miR-146a, miR-223, and miR-483, while the frail ones showed higher expression of miR-21, miR-223, and miR-483 when compared to the control group. However, the authors found that miR-223 and miR-483 levels increased to a similar extent in robust and frail individuals matched by age, so both biomarkers should be considered as biomarkers of aging but not frailty. miR-223-5p targets BMI1, a transcript of a key gene involved in the self-renewal of bone marrow mesenchymal stem cells playing a critical role in promoting osteogenesis [84]. Further studies with bigger cohorts are needed to explore the potential of miR-223 as a candidate frailty biomarker due to its association with bone mass loss and osteoporosis [85].

Rusanova and coworkers have also reviewed different families of microRNAs linked to “inflammaging” (inflamma-miRs), to musculoskeletal health (myomiRs), and microRNAs that can directly or indirectly affect the mitochondrial function (mitomiRs) [69]. The list of miRNAs that can be considered as frailty biomarkers include: miR-1, miR-21, miR-34a, miR-146a, miR-185, and miR-206, miR-223 [69]. The importance of miR-21 as a potential frailty biomarker has also been suggested by other research groups [86].

By using muscle biopsies and a microarray-based experimental approach, Zheng et al. found that miR-34a-5p and miR-449b-5p levels were elevated in sarcopenic muscles, highlighting their importance in muscle aging [87].

Further studies should be performed to increase the number of subjects and to provide reliable values for sensitivity and specificity, which may increase the validity of these miRNA as frailty biomarkers.

Regarding lncRNAs, in the LonGenity study, genome-wide association studies were used to explore whether variations in the 9p21–23 locus played a role in frailty in 637 community-dwelling older individuals [88]. The authors found associations between SNPs in the regulatory 9p21–23 region and the frailty phenotype; signifying the importance of this locus in aging [88]. Interestingly, the genomic locus 9p23 harbors several genes including ANRIL, a long non-coding RNA gene associated with cardiovascular diseases and strokes [89]. A polymorphism rs2811712 located in ANRIL gene has been associated with physical function in older people (65–80 years) with the minor allele being associated with reduced physical impairment [90]. However, although this SNP located into lncRNAs was indirectly associated with physical impairment, its expression levels have not yet been explored.

Other lncRNAs are feasible candidates to be investigated in elderly people to evaluate its contribution to frailty [91]. In this regard, lncRNA H19, is an interesting candidate because it plays a key role in myoblast differentiation during skeletal muscle regeneration by negatively regulating the bone morphogenetic protein (BMP) pathway [92]. Other lncRNAs that are related with muscle cells differentiation and proliferation are MALAT1, linc-MD1, and SIRT1 AS. lncRNA MALAT1 is downregulated by myostatin [93] and it suppresses the proliferation of myoblasts [94], suggesting that it may influence myogenesis during aging. Regarding linc-MD1, it is involved in the decline in skeletal muscle regeneration via HuR [95], a gene which is downregulated in differentiated muscle cells and contributes to sarcopenia. Another interesting lncRNA is SIRT1 AS, which is a natural antisense of the NAD-dependent deacetylase Sirt1. SIRT1 AS controls myogenic programs during muscle aging [96].

### 3.3. Epigenetic Biomarkers to Follow-up the Interventions in Frailty

There are two important characteristics regarding the frailty syndrome, firstly, if left unaddressed it will evolve into disability and eventually, death, and secondly, if it is properly treated, its onset can be delayed and the physiological condition of the subject can be improved [97].

The implementation of intervention programs in the elderly is not an easy task since it is a very heterogeneous population [98]. This heterogeneity is the basis of the previously mentioned field of personalized medicine, whose objective is to adapt interventions and treatments individually, considering the patient’s lifestyle [99]. Interventional studies would undoubtedly benefit from the identification of frailty biomarkers, including epigenetic biomarkers.

There are no known pharmacologic interventions for the prevention of frailty [100]. However, because of major advances in understanding the molecular basis of aging, there is now tremendous interest and it is a very active area of investigation in the search for agents that may potentially modify human aging and health span. Delaying the onset of frailty is tightly correlated with improvement in health span. Although a single definition of health span is not available, a common definition is: “the period of life spent in good health, free from the chronic diseases and disabilities of aging” [101]. Several anti-aging interventions have potential translatability in the treatment and prevention of frailty [102]. Some of these interventions have been very well characterized in animal studies but they need to be tested in humans. The main pharmacological interventions in aging with potential translation to frailty are: caloric restriction mimetics such as (i) metformin, (ii) rapamycin, and (iii) resveratrol as well as more novel approaches that are emerging in the field such as (iv) nicotinamide adenine dinucleotide precursors, (v) synthetic activators of sirtuins such as SRT2104, and (vi) senolytics (dasatinib and quercetin) [102]. The ENRGISE (Enabling Reduction of Low-Grade Inflammation in Seniors) study deserves a comment in this section because it is an ongoing clinical trial (NCT02676466) that is examining the effect of fish oil and angiotensin receptor blockade on systemic inflammation and gait speed [100].

The main interventions developed to date to improve frailty-related health outcomes include lifestyle/behavioral factors: exercise, nutrition, multicomponent interventions, individually-tailored geriatric care models and cognitive health maintenance [32,100]. Among them, exercise is considered the most effective intervention in preclinical and clinical models of frailty [103,104].

In clinical practice, it has been shown that frailty can not only be delayed but also reversed by exercise training [105]. The use of an appropriate exercise program can delay or even reverse the physiological changes related to age that occur at the musculoskeletal level [97,106]. Multicomponent interventions have also proved beneficial to treat frailty. Multicomponent exercise is defined as a program of endurance, strength, coordination, balance, and flexibility exercises that have the potential to impact a variety of functional performance measurements. This type of exercise is a recommended alternative to more traditional exercise regimens, particularly due to its potential to impact functional performance outcomes in older adults [104,107,108,109,110,111,112,113,114,115].

One of the most successful multi-center intervention clinical trials with a one-year supervised physical activity program, “Lifestyle Interventions and Independence for Elders Pilot (LIFE-P) Study” (NCT00116194), established a 20% reduction in the prevalence of at least one criterion of frailty one year after the exercise program [116,117].

Exercise has an impact on several of the root mechanisms of aging also known as biological aging [118]. By doing so, it can delay phenotypic aging.

Despite the evidence of the benefits of physical exercise on health status, the prevalence of inactivity (35%) in subjects aged ≥75 years is worrisome [119]. Therefore, is very important to promote physical activity in this group of people. Several research groups have shown that supervised intervention programs aimed at improving the functionality of the elderly, fundamentally based on physical exercise, are more effective than those carried out autonomously in the subject’s environment [109].

Physical exercise can influence fundamental epigenetic mechanisms such as DNA methylation in skeletal muscle. DNA methylation not only tracks chronological age in humans but also phenotypic changes along lifespan, predicting, for instance, cardiovascular mortality and other age-associated adverse outcomes [45]. Exercise-associated decrease in whole genome methylation has been found in muscle biopsies in healthy sedentary individuals after an acute bout of exercise [120,121]. This hypomethylation in specific promoters was accompanied by an increased expression of some genes involved in energy metabolism and mitochondrial function. These results suggest that DNA methylation changes can represent an active and adaptive response to skeletal muscle contraction. Methylation changes have also been analyzed in human skeletal muscle after a training program [122]. In this study the authors reported a significant modulation in DNA methylation in genes involved in structural changes of muscle tissue, inflammation, and immunological pathways after three months of endurance exercise training.

Lifelong physical activity is also able to induce hypomethylation in promoters of genes involved in resistance to oxidative stress, energy metabolism or myogenesis [123]. The exercise-induced epigenetic changes can be retained, and that DNA methylation could underpin the capacity of skeletal muscle to maintain information into later life and to respond differently to previous stimuli such as training [123].

Our current knowledge on how age-associated DNA methylation changes are related to frailty and the role of interventions such as exercise in epigenetic modifications is still sparse. Further research is needed because all the studies published up until now have been performed on young and healthy adults, and to date, no interventional studies have examined the effects of training on DNA methylation status in elderly people.

Apart from physical exercise, malnutrition and loneliness are other key aspects in the functional deterioration of the elderly. The prevalence of malnutrition in Western Europe in people over 65 is 23% on average and ranges between 6% and 51% [7]. This malnutrition, produced by a deficit of calories or protein, must be considered when proposing interventions aimed at improving the quality of life of our elderly.

## 4. Discussion and Conclusions

The identification of epigenetic biomarkers may contribute to the early detection of subclinical changes or deficits at the molecular and/or cellular level, to prevent or delay the development of frailty, and also its consequences. This will provide important support for the clinical diagnosis of frailty or any of its associated risks. Interventional studies would undoubtedly benefit from the search for biomarkers of frailty.

In addition, the identification of biomarkers can help to improve the health trajectory of the elderly, postponing and mitigating the appearance of functional problems. Improving the clinical follow-up of the elderly as well as predicting the future evolution of frailty and dependency in these people is a social and medical commitment. In this regard, it is necessary to orient research efforts in the design of personalized pharmacological and non-pharmacological interventions as a good example of personalized medicine.

## Figures and Tables

**Figure 1 ijerph-18-01883-f001:**
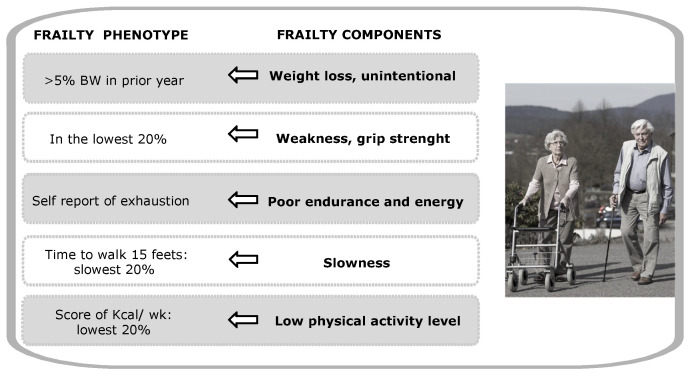
Frailty features contributing to the frailty phenotype during aging.

**Table 1 ijerph-18-01883-t001:** Epigenetic biomarkers to study frailty-related phenotypes.

Epigenetic Mechanism	Epigenetic Biomarker Associated to Frailty	Biospecimen	Association between the Epigenetic Change and Frailty	Reference
DNA methylation	Horvath’s clock	Leukocytes	Epigenetic age acceleration was correlated with clinically relevant aging-related phenotypes	[54]
Histone PTMs	H3K9me3 and H3K9ac and H3K27ac decreases with age	Muscle samples in rat models	Muscle loss and sarcopenia	[79]
	γH2AX	Leukocytes	Significantly higher γH2AX values observed in individuals positive for low physical activity (*p* < 0.001), slow waking (*p* < 0.01), and low grip strength (*p* < 0.01) criteria	[81]
Non-coding RNAs	miR-10a-3p, miR-92a-3p, miR-185–3p, miR-194–5p, miR-326, miR-532–5p, miR-576–5p, miR-760	Plasma exosome-derived miRNAs	Associated to frailty	[82]
	miR-21, miR-223, miR-483	Plasma	Increased expression in frail subjects compared to control subjects	[83]
	miR-146a	Plasma	Low levels in frail subjects compared to robust old adults	[83]
	miR-1, miR-21, miR-34a, miR-146a, miR-185, miR-206, miR-223	Plasma	Increased levels in frail subjects compared to robust old adults	[69]
	miR-34a-5p miR-449b-5p	Muscle biopsy	Elevated in sarcopenic muscle compared with muscle tissue from controls	[87]

## Data Availability

Not applicable.

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
