# Peer review of "Implementing Precision Medicine in Human Frailty through Epigenetic Biomarkers"

_ijerph, 2021, doi:10.3390/ijerph18041883_

Round 1

Reviewer 1 Report

It was a pleasure for me to read the review

Implementing precision medicine in human frailty through ep-igenetic biomarkers.

by José Luis García-Giméneza et al.

It is clearly written, well-structured and it addresses virtually all relevant points related to the topic.

Minor points:

  1. Line 167: please check spelling: are review(ed)?
  2. Line 313: subsection 3.2 instead of 3.1
  3. Line 479: cite PMID
  4. What does ‘global H3-methylation increase’ mean? At all side chains, K4, K9, K27, K36 etc? The same applies to acetylation. Please specify and/or comment.

Author Response

 ANSWERS TO REVIEWERS

[IJERPH] Manuscript ID: ijerph-1075292 

Reviewer #1

It was a pleasure for me to read the review: Implementing precision medicine in human frailty through ep-igenetic biomarkers by José Luis García-Giméneza et al.It is clearly written, well-structured and it addresses virtually all relevant points related to the topic.

Answer: We thank reviewer 1 for his/her valuable evaluation and encouraging comments. We have considered his/her minor recommendations and suggestions. Itemized responses are listed below. All the modifications have been marked in red throughout the manuscript to make its revision easier.

  1. Line 167: please check spelling: are review(ed)?

Answer: We have corrected the spelling. 

  1. Line 313: subsection 3.2 instead of 3.1

Answer: The subsection numbering has been corrected in several parts of the review. We thank the reviewer for noticing this error.

  1. Line 479: cite PMID

Answer: We have corrected this mistake and included this reference in its proper format.

  1. What does ‘global H3-methylation increase’ mean? At all side chains, K4, K9, K27, K36 etc? The same applies to acetylation. Please specify and/or comment.

Answer: When we indicate global changes, we are referring to changes in H3 throughout the whole genome in a specific tissue instead of a local change in the chromatin associated to a specific gene. Following the reviewer’s comment, we have modified Table 1 to clarify the specific histone modifications that has been reported in the literature.

Reviewer 2 Report

The manuscript is focusing on the epigenetic biomarkers related to the human frailty. Authors described the frailty and aging and reviewed various epigenetic studies associated with aging and frailty. In overall, the review is well written and interesting, but I would like to point out several points for the manuscript.

  • In the 3.1.3, authors described the non-coding RNAs and their roles in the gene regulation. Also, the role of miRNAs, one of the non-coding RNA in aging have been described in the section. But authors did not mention about the long-noncoding RNAs in aging area. Is there any study about the relationship between long-non coding RNAs and aging?

  • Authors listed the epigenetic biomarkers to study frailty-related phenotypes in Table 1. Related to the DNA methylation mechanism, authors listed only one clock. But in the previous section, “3.1.1. DNA methylation”, authors introduced various epigenetic clocks associated with epigenetic age and age-related outcomes. Why were those clocks not listed in the Table 1?

  • There are typos in the manuscript like below. Authors should carefully check those typos and mistakes (punctuation marks, spaces, parenthesis, etc…) in the whole manuscript.

  • There are errors in the numbering of the following sections.
  • In page 7, line 313: “3.1. Epigenetic biomarkers associated to aging and frailty” 3.1 => 3.2
  • In page 10, line 432: “3.2. Epigenetic biomarkers to follow-up the interventions in frailty”3.2 => 3.3

Author Response

ANSWERS TO REVIEWERS

[IJERPH] Manuscript ID: ijerph-1075292

Reviewer #2

The manuscript is focusing on the epigenetic biomarkers related to the human frailty. Authors described the frailty and aging and reviewed various epigenetic studies associated with aging and frailty. In overall, the review is well written and interesting, but I would like to point out several points for the manuscript.

Answer: We thank reviewer 2 for his/her evaluation. Itemized responses are listed below. All the modifications have been marked in red throughout the manuscript to make its revision easier.

  1. In the 3.1.3, authors described the non-coding RNAs and their roles in the gene regulation. Also, the role of miRNAs, one of the non-coding RNA in aging have been described in the section. But authors did not mention about the long-noncoding RNAs in aging area. Is there any study about the relationship between long-non coding RNAs and aging?

Answer:  We did not mention long non-coding RNAs in the first version of our manuscript because there is scarce evidence describing its role in frailty. Nonetheless, after careful review of the literature we have included a new section in the paper to provide the most relevant information regarding the potential role of lncRNAs in frailty.

  1. Authors listed the epigenetic biomarkers to study frailty-related phenotypes in Table 1. Related to the DNA methylation mechanism, authors listed only one clock. But in the previous section, “3.1.1. DNA methylation”, authors introduced various epigenetic clocks associated with epigenetic age and age-related outcomes. Why were those clocks not listed in the Table 1?

Answer: We have only included the Horvath’s clock in Table 1 because it is the only one that has been used to study epigenetic changes and its correlation with age-related phenotypes, such as the frailty phenotype. The manuscript cited in table 1 reports that frailty is associated with the epigenetic clock but not with telomere length in a German cohort [1]. No other clocks have been used to explore the relation between epigenetic age acceleration and frailty. Further studies exploring the association of these new clocks with frailty are needed to demonstrate their usefulness to assess the relationship between epigenetic age acceleration and frailty-related phenotype.

3.There are typos in the manuscript like below. Authors should carefully check those typos and mistakes (punctuation marks, spaces, parenthesis, etc…) in the whole manuscript.

    • There are errors in the numbering of the following sections.
    • In page 7, line 313: “3.1. Epigenetic biomarkers associated to aging and frailty” 3.1 => 3.2
    • In page 10, line 432: “3.2. Epigenetic biomarkers to follow-up the interventions in frailty”3.2 => 3.3

Answer: We thank the reviewer for noticing these errors. We have corrected them in the new version of the manuscript.

References:

  1. Breitling, L.P., et al., Frailty is associated with the epigenetic clock but not with telomere length in a German cohort. Clinical Epigenetics, 2016. 8(1): p. 21-21.

Reviewer 3 Report

The manuscript by Garcia-Gimenez et al describes the state of the art on the epigenetic mechanisms and biomarkers associated with frailty.

The manuscript is interesting and well-witten.

Only few issues need to be addressed:

1) page 4 lane 166-174: please eliminate the description of the content of the sections, since it is not needed in a linear manuscript

2)page 7 lane 283-284: again eliminate the description of a section of the manuscript

4)page 9, lane 418: correct and after Rusanova

5)Page 9, lane 422: adda aspece between of and miR-21

6)page 12 lane 528-534: this conclusion is repetitive. Please, eliminate it or substitute it with the future perspectives.

Author Response

ANSWERS TO REVIEWERS

[IJERPH] Manuscript ID: ijerph-1075292

Reviewer #3

Comments and Suggestions for Authors

The manuscript by Garcia-Gimenez et al describes the state of the art on the epigenetic mechanisms and biomarkers associated with frailty. The manuscript is interesting and well-witten. Only few issues need to be addressed:

Answer: We thank reviewer 3 for his/her valuable comments. Itemized responses are listed below. All the modifications have been marked in red throughout the manuscript to make its revision easier.

1.Page 4 lane 166-174: please eliminate the description of the content of the sections, since it is not needed in a linear manuscript

2)page 7 lane 283-284: again eliminate the description of a section of the manuscript

Answer: Following the reviewer’s advice we have eliminated both paragraphs in the new version of the manuscript.

4)page 9, lane 418: correct and after Rusanova

5)Page 9, lane 422: adda aspece between of and miR-21

Answer: We have corrected both typos. 

6)page 12 lane 528-534: this conclusion is repetitive. Please, eliminate it or substitute it with the future perspectives.

Answer: We have eliminated this part of the conclusion section of the manuscript.